# Development of Field Tests for Cardiovascular Fitness Assessment in Wheelchair

**DOI:** 10.3390/healthcare12050580

**Published:** 2024-03-02

**Authors:** Eun Hyung Cho, Bong-Arm Choi, Yongsuk Seo

**Affiliations:** 1Korea Institute of Sport Science, Seoul 01794, Republic of Korea; ehcho@kspo.or.kr; 2Department of Physical Education, Daegu University, Daegu 38453, Republic of Korea; ba3665@naver.com

**Keywords:** wheelchair athletes, cardiovascular fitness, field tests

## Abstract

It is essential to consider both physique and physical fitness factors to minimize the risk of injuries and optimize athletic performance among elite athletes. Athletes with disabilities face limitations in fitness assessments compared to their healthy counterparts. The aim of this study was to revalidate established cardiovascular fitness assessment methods and develop field tests for wheelchair athletes. As representatives registered at the Korea Paralympic Committee’s Athletes Training Center in Icheon, athletes with physical disabilities participating in para ice hockey (n = 14), who were capable of wheelchair control, were volunteered. Prior to cardiovascular fitness assessments using an ergometer and a shuttle run, demographic characteristics were surveyed, and physical measurements and muscle strength (grip strength) were recorded. All the participants performed one ergometer test based on cardiovascular fitness criteria, and for shuttle run validation, two trials were conducted using existing audio cues (National Physical Fitness 100, 20 m shuttle run). For the development of the shuttle run, considering wheelchair turning, signal-to-sound intervals were increased by 1 s and 1.5 s, respectively, in two trials. An analysis of the correlation with the maximal oxygen consumption (VO_2_max) in comparison to the reference criterion, an ergometer, demonstrated high correlations in the first trial (r = 0.738) and the second trial (r = 0.780). Similarly, significant correlations were observed with the maximum heart rate (HRmax) in the first trial (r = 0.689) and the second trial (r = 0.896). Thus, the 15 m shuttle run is validated as a field test for assessing cardiovascular fitness in athletes with disabilities. Correlation analysis with maximal oxygen uptake (VO_2_max) compared to the reference criterion, an ergometer, revealed a correlation of 0.815 with a 1 s interval audio cue and 0.355 with a 1.5 s interval audio cue. A high correlation was observed with the 1 s interval audio cue. Regarding the maximum heart rate (HRmax), the correlations were 0.665 with a 1 s interval audio cue. Once again, a high correlation was noted with the 1 s interval audio cue. The field test selected for measuring cardiovascular fitness in wheelchair athletes involved performing a 15 m shuttle run while in the wheelchair. The test utilized an audio cue with a 1 s increased interval between the signal sounds.

## 1. Introduction

It is essential to consider not only the physique of athletes but also the physical factors required for the specific sport to minimize the risk of injuries and optimize the individual capabilities of elite athletes for peak performance [1]. Elite sports athletes should undergo training and competition based on accurate diagnoses of both physique and physical fitness factors through suitable measurement and evaluation tools [2]. Specifically, for athletes with disabilities, limitations in the range of physical fitness factors arise due to both physical and mental impairments when compared to their healthy counterparts [3]. Consequently, it is imperative for athletes with disabilities to undergo precise fitness assessments as a fundamental prerequisite before training and competition [4]. Based on the assessments, it is crucial to modify interventions to the specific characteristics of the disability type and sports discipline. This approach aims to enhance motor functions and physical fitness, ultimately facilitating the proficient application of sports techniques. Cardiorespiratory fitness, among various components of physical fitness, is closely associated with exercise and is considered a fundamental element in fitness assessment. Methods for assessing cardiorespiratory fitness include conducting exercise stress tests using measurement equipment such as a treadmill, cycle ergometer, or arm ergometer while simultaneously performing respiratory gas analysis to directly measure the subject’s cardiorespiratory fitness. Alternatively, indirect estimation methods involve conducting field tests, like the shuttle run, the step test, or endurance running, using measured parameters such as heart rate and exercise intensity. Direct measurement methods, conducted by specialized personnel using equipment, offer relatively accurate results but have the drawbacks of requiring expensive equipment, skilled personnel, and consuming a significant amount of time for the assessment [5]. Indirect measurement methods are advantageous in that they are conducted in the field environment where competitions and training take place. This makes the measurements less accessible, but it provides a familiar setting for the subjects, promoting psychological stability and motivating their participation in the tests [6]. On the other hand, when using indirect measurement methods, reliance on correlation equations for estimation introduces a drawback of lower reliability compared to direct measurements. Individuals living with wheelchair mobility typically have less than 50% of the muscle mass used in daily activities compared to the general population, resulting in lower cardiorespiratory fitness and energy efficiency. Additionally, as they use their arms and shoulders for nearly all daily activities, the upper limb muscles and the skeletal muscles in the upper extremities are more prone to fatigue. Therefore, cardiorespiratory fitness and strength in the upper limbs are crucial for individuals using wheelchairs [7].

Moreover, incorrect exercise for wheelchair users can pose a risk factor for adverse effects on the cardiovascular and musculoskeletal systems. Therefore, accurately assessing the pre-exercise fitness level is crucial for wheelchair users, taking into consideration factors such as the nature of the disability, the classification, and the specific goals [8]. Currently, direct measurement methods for assessing cardiorespiratory fitness in wheelchair users involve exercise stress tests using upper limb-powered devices like the arm ergometer and the wheelchair treadmill, coupled with simultaneous respiratory gas analysis to directly measure cardiorespiratory fitness. Previous studies reported that indirect measurement methods involve field tests conducted in the wheelchair exercise environment, where athletes traverse a specified distance or time [9,10,11]. 

Field tests commonly used for assessing the cardiorespiratory fitness of wheelchair users include the multistage field test, the modified Cooper test, the 25 m shuttle run test, the arm ergometer test, and the 12 m wheelchair shuttle run test [6,9,12,13,14,15]. However, wheelchair users encounter limitations in measurements due to their distinct physical characteristics compared to the general population. These challenges arise from adaptations of tests originally designed for able-bodied individuals, leading to the necessity for modifications for wheelchair users. Therefore, there is a need for the development of field tests that are relatively less restrictive and can be conducted by athletes or coaches in the exercise environment rather than experimental tests. The purpose of the current study was to develop a field test for assessing cardiovascular fitness in wheelchair athletes, aiming for a more convenient method to measure a larger number of participants. We intended to modify the shuttle run, a test currently used to assess cardiovascular fitness, to suit athletes using wheelchairs, thereby developing a modified version called the shuttle ride. To achieve this, we revalidated the existing measurement tools currently in use. 

## 2. Materials and Methods

### 2.1. Research Subjects

The study targeted athletes with disabilities registered as national representatives in the Korea Paralympic Committee’s Athletes Training Center in Icheon, specifically those with lower limb impairments who were capable of wheelchair control in wheelchair basketball and para ice hockey. Before initiating this research, an orientation session was conducted to explain the purpose, measurement items, and schedule of the study. Following this, informed consent was obtained from the participants who decided to take part in this study. A total of 40 participants were recruited in consultation with the Training and Education Department of the Icheon Athletes Training Center. Some individuals dropped out during the measurement process due to injury, condition issues, voluntary withdrawal, etc., and the final measurements were conducted on 14 para ice hockey players (age 38.9 ± 8.8 years; weight 75.9 ± 12.4; height, 169.3 ± 13.1) and 13 wheelchair basketball players (age 33.7 ± 7.3 years; weight 74.3 ± 13.5l; height, 177.7 ± 5.7). Participants were excluded if they experienced difficulties performing exercises using their hands and arms, operating a wheelchair, or engaging in arm ergometer exercises. Additionally, individuals with challenges in conducting exercise stress tests due to cardiovascular conditions, respiratory symptoms associated with infectious diseases, or any other physical or psychological difficulties were excluded. Furthermore, voluntary withdrawal from research participation was also considered a criterion for exclusion. 

The experiments were divided into separate groups to enable better control over the potential confounding variables associated with training regimens, coaching styles, and injury prevalence.

The types and causes of the disabilities among the participants in each sport are detailed in Table 1. The participants were involved in the study for a duration of 5 days, with measurements conducted three times (on days 1, 3, and 5). The protocol was approved by the Korea Institute of Sport Science Institutional Review Board (IRB No. KISS-21027-2109-02) and all the volunteers provided written informed consent. 

### 2.2. Procedure 

#### 2.2.1. Familiarization Session

On the first visit to the laboratory, personal information such as sports discipline, experience, and anthropometry data (age, weight, height) and disability-related details (type, location, classification, cause, duration) was obtained from each participant (Table 2). The participants chose wheelchairs with suitable seat widths for exercising. Adequate time was provided for them to become familiar with standard wheelchairs before the measurements. Specifically, practice sessions were provided for rotating the cone.

#### 2.2.2. Incremental Arm Ergometer Test 

For the evaluation and development of the cardiorespiratory fitness test, the participants underwent one session of the arm ergometer test for the criterion. Each athlete underwent an incremental arm cranking exercise (ACE) test using an isopower ergometer (ER-800; Ergoline, Bitz, Germany) to determine the maximal oxygen uptake (VO_2_max) under standardized laboratory conditions.

VO_2_ and HR were measured using a portable metabolic system (K5, COSMED, Rome, Italy) and a heart rate monitor (Polar RS800 CS, Polar Electro Oy, Kemple, Finland), respectively (Figure 1). While seated in a chair or wheelchair, the ergometer crankshaft and shoulders were adjusted to align, ensuring a comfortable position for exercise. Prior to the measurements, a sufficient warm-up period (approximately 5 min) was provided. The arm ergometer test protocol began at an initial load of 25 Watts, with increments of 15 Watts every 2 min. The crank rotation speed was maintained at 50–60 rpm. The criteria for terminating the exercise were when the subject could no longer sustain the activity or when the crank rotation speed dropped below 49 rpm. After exercise cessation, the exercise duration and RPE were recorded.

#### 2.2.3. Experiment 1 Procedure

To validate the existing field test, 14 para ice hockey players performed two 15 m shuttle ride sessions using the traditional audio cues. The protocol of the shuttle ride is shown in Table 3.

In this study, since the participants were a specialized group of athletes with disabilities, we consulted previous research. The arm ergometer was set to the maximum exercise intensity level, while the shuttle ride distance was adjusted from the standard 20 m used for the general population to 15 m. Additionally, the elimination criteria were relaxed, allowing the participants to reach within a 3 m range from the line instead of strictly requiring arrival at the finish line.

#### 2.2.4. Experiment 2 Procedure

For the development of the field test, considering the wheelchair turning time at the starting and finishing lines, two shuttle run sessions were conducted using audio cues with 1 s (1st shuttle ride) and 1.5 s (2nd shuttle ride), respectively (Table 3). The order of the shuttle rides was counterbalanced. 

#### 2.2.5. Instrumentation 

The participants wore a portable metabolic system for measurement of oxygen uptake and a chest band for heart rate measurement (respiratory gas and heart rate were continuously analyzed during shuttle rides). The participants selected wheelchairs with an appropriate seat width for performing the exercise. Sufficient time was allowed for the participants to familiarize themselves with the standard wheelchair, including practicing turns.

The Delphi survey method was employed to gather expert opinions and achieve a consensus on the crucial factors influencing the development of the field test for assessing cardiovascular fitness among wheelchair-using athletes during shuttle rides [16]. The starting and finishing lines were positioned 15 m apart, with cones placed along the line to facilitate wheelchair rotation. Additionally, a 3 m allowance zone was designated inside each line, with guidelines marked to assist the participants in orientation. The shuttle ride proceeded in synchronization with an audio cue, including signals for the start and speed changes. The test was terminated if, on two or more occasions, the participant was more than 3 m away from the line at the time of the signal (failure to enter the allowance zone). During the shuttle ride test, VO_2_ and HR were continuously monitored and stored (Figure 2).

### 2.3. Statistical Analyses

All the measurement data were calculated as the mean and standard deviation. To validate the reliability of the arm ergometer and the shuttle ride test, correlation analysis was conducted along with the presentation of Bland–Altman plots., Another correlation analysis was conducted between the arm ergometer and the modified shuttle ride test. Also, the VO_2_max and HRmax data during arm ergometer were analyzed between two different sports with independent samples *t*-tests. The statistical significance level was set at α = 0.05 for all analyses.

## 3. Results

### 3.1. Inter-Sport Comparison of Cardiorespiratory Fitness with Arm Ergometer

The independent sample *t*-tests showed no statistically significant differences in the VO_2_max (t = 0.485, *p* = 0.633), with a small effect size of Cohen’s d = 0.18. Similarly, for the HRmax, there were no significant differences (t = −0.185, *p* = 0.854), with an effect size of d = 0.07, also indicating a small effect [17]. The average VO_2_max was 35.9 ± 8.9 and 34.6 ± 4.5 mL/kg/min in the para ice hockey and wheelchair basketball, respectively. Furthermore, the average HRmax was 170.6 ± 18.2 and 172.0 ± 14.36 beats/min in the para ice hockey and wheelchair basketball, respectively. 

### 3.2. Validity of Shuttle Ride 

A simple linear regression analysis for the VO_2_max between the arm ergometer and the shuttle ride indicated a significant positive relationship for the 1st shuttle ride (r = 0.738, *p* = 0.003) with a medium effect size of Cohen’s d = 0.73 and the 2nd shuttle ride (r = 0.780, *p* = 0.001) with a medium effect size of Cohen’s d = 0.77 [17]. The Bland–Altman results presented in Figure 3 show the relationship between the VO_2_max of the AE and the SR. The Bland-Altman plots indicate a bias of −5.4 (LoA: +6.51, −17.2) for the 1st SR and −3.3 (LoA: +8.2, −14.7) for the 2nd SR. 

A simple linear regression analysis (Figure 4) for the HRmax between the arm ergometer and the shuttle ride indicated a significant positive relationship for the 1st shuttle ride (r = 0.689, *p* = 0.006) and the 2nd shuttle ride (r = 0.896, *p* ≤ 0.001).

### 3.3. Correlation between Modified Shuttle Ride and Arm Ergometer

A simple linear regression analysis for the VO_2_max between the arm ergometer and the modified shuttle ride indicated a significant positive relationship for the modified shuttle ride (+1 s) (r = 0.815, *p* = 0.001). The Bland–Altman plots indicate a bias of −1.1 (LoA: +4.6, −6.8) in Figure 5.

## 4. Discussion

This study was conducted with the aim of developing a field test for measuring the cardiovascular fitness of athletes who use wheelchairs. To accomplish this objective, a literature review and expert meetings were conducted to choose the measurement tools and variables for the assessment of cardiovascular fitness. The selected tools were then validated and developed by applying them to nationally representative athletes with disabilities.

For athletes using wheelchairs, lower limb functionality is replaced by the wheelchair due to lower limb dysfunction, paralysis, or loss [18]. These athletes have characteristics such as increased mobility and activity through wheelchair use. Additionally, physiological responses and development may differ from athletes who use both upper and lower limbs due to limitations in lower limb movement [19]. Therefore, applying tests designed for healthy athletes directly to those using wheelchairs is challenging [20]. One fundamental and widely used variable for evaluating cardiovascular fitness is maximal oxygen consumption (VO_2_max), which can be measured through laboratory tests and field tests. Field tests, particularly those using incremental load methods by adjusting time and speed, bring out the maximum cardiovascular capacity of the test subjects [21].

In the case of field tests for assessing cardiovascular fitness, activities such as shuttle runs and Harvard step tests often involve lower limb movements, making them unsuitable for athletes with lower limb disabilities. As a result, ergometers such as arm ergometers and wheelchair treadmills have been employed in laboratory settings to gauge cardiovascular fitness. Nevertheless, athletes encountered constraints and fewer measurement opportunities with these methods, underscoring the necessity for field test measurement tools.

Internationally, examples of field tests for evaluating cardiovascular fitness in wheelchair users include the continuous multistage field test, which measures the VO_2_max by incrementally increasing the exercise intensity indoors [10]. This test is conducted in a field where actual training and competitions take place, providing data related to real game performance. Another example is the intermittent fitness test, which includes intermittent rest periods. However, some studies found no statistical significance between continuous and intermittent tests, while others reported the effectiveness of intermittent tests in studies applying maximum load exercises [22,23].

Previous research has been conducted on wheelchair basketball players using shuttle run tests to estimate the VO_2_max and determine the optimal shuttle run distance. The results indicated that a 14 m distance had the highest explanatory power for estimating the VO_2_max (83.4%, R^2^ = 0.834) [24]. Additionally, the 15m shuttle run test showed high validity and reliability in measuring aerobic capacity in individuals with cerebral palsy who experience movement disorders due to muscle spasticity and contracture [25,26]. A previous study developed a 25 m indoor “shuttle run” test incorporating auditory feedback signals to evaluate the aerobic capacity of seasoned wheelchair basketball players [27]. The authors validated their assessment by contrasting the maximal heart rates (HRs) recorded during the field test with those obtained during an arm-crank laboratory test, yielding a correlation coefficient of r = 0.78. Another previous study undertook a comparison of the maximal cardiorespiratory variables observed during an incremental field test on a 400 m tartan track with those measured in a controlled laboratory environment [6]. The authors found a moderate correlation coefficient of r = 0.65 between the peak VO_2_ values obtained from the two trials. Also, Vanderthommen et al. developed a simple indoor multistage field test and assessed the reliability of the multistage field test. The results indicated that the 180° turns after each 25 m and the resulting cumulative decelerations and accelerations of the wheelchair-user system probably solicited anaerobic energy sources to some extent. This was less critical with the present approach, where the participants wheeled around an octagonal track. The changes in direction were less abrupt, thus avoiding high decelerations and energy losses during the turns. Indeed, during the final exercise stage, the subjects lost only about 25 percent of their average velocity maintained during that level [10]. 

The validity of the 15 m shuttle ride as a cardiovascular fitness measurement tool was confirmed in this study. Similar to previous research, high correlation coefficients were observed between the VO_2_max and the HRmax measured during two rounds of the 15 m shuttle ride and those measured using the arm ergometer. The study also considered the wheelchair rotation time during turns at the starting and finishing lines to ensure that wheelchair control did not impact the measurement variables. The results showed that increasing the duration of the auditory cue by 1 s made the test more suitable for measuring cardiovascular fitness. Considering the results of this study and previous research, the 15 m shuttle ride is a suitable measurement tool for assessing cardiovascular fitness in athletes using wheelchairs. 

### Limitations

This study has several limitations that warrant consideration for generalization and future research. While it is advisable to allow a 3 to 4-day rest period between assessments to mitigate the influence of fatigue [28], practical constraints in the research schedule and athlete availability necessitated a 1-day rest period between measurements. The participants were advised against engaging in intense training or personal workouts during this period to ensure sufficient rest. Moving forward, for studies aiming to develop field tests for assessing cardiovascular fitness in wheelchair athletes, it is recommended to explore methods that account for wheelchair skill’s potential impact on the measurement outcomes. Additionally, comparing wheelchair users with healthy individuals could establish baseline data for the observed differences, while investigating the influence of the wheelchair type on sensory perception and functionality may offer insights into enhancing exercise capabilities [29].

Moreover, there is a need for the development of wearable devices specifically designed to measure physical activity and other functions in wheelchair athletes. Such devices could incorporate features customized to their characteristics, facilitating more efficient and accurate measurements during exercise. Enhancements could also involve integrating additional metrics, such as the maximal rolling velocity, to assess exercise performance alongside existing measurement factors. Furthermore, evaluating metabolic rates, muscle mass, and exercise capabilities based on age and sex among wheelchair athletes is recommended. Comparative analyses of physiological and physical characteristics across various age groups could provide valuable insights into age-related differences, while efforts to refine measurement tools to accurately reflect the unique characteristics of disabilities in athletes are essential for more precise fitness assessments. Adjusting measurements or developing specialized methods considering individual variations in disability severity, duration, and assistive device usage could further improve assessment accuracy.

## 5. Conclusions

This study sought to establish a field test for evaluating cardiovascular fitness in athletes with disabilities. The findings, derived from testing on nationally representative athletes, reveal that the 15 m shuttle ride proves to be a suitable tool for assessing cardiovascular fitness in wheelchair athletes when validated against reference-based arm ergometry. Notably, the ultimate selection of an audio cue featuring a 1 s increase from the original shuttle run audio cue demonstrated stronger correlations in both the VO2max and HRmax.

## Figures and Tables

**Figure 1 healthcare-12-00580-f001:**
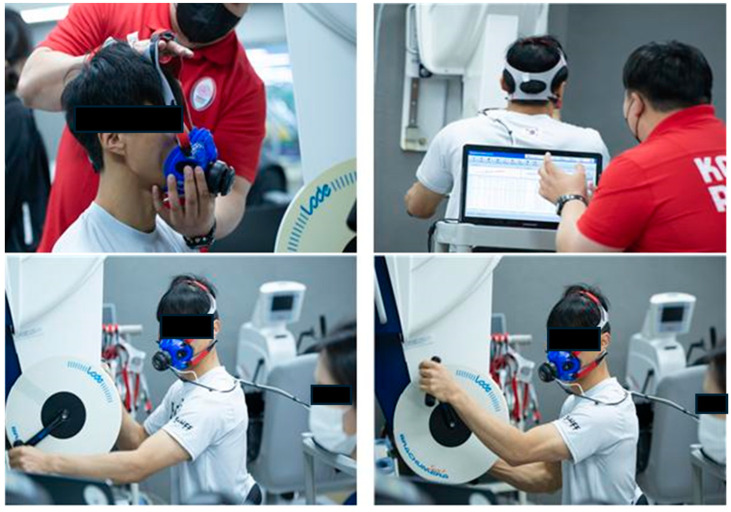
Arm ergometer test.

**Figure 2 healthcare-12-00580-f002:**
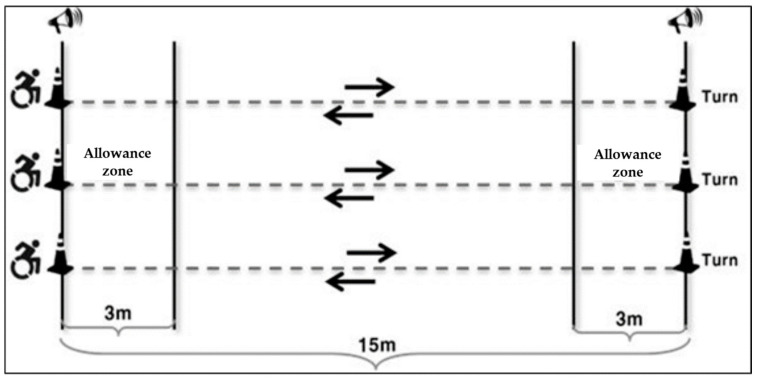
The shuttle ride process.

**Figure 3 healthcare-12-00580-f003:**
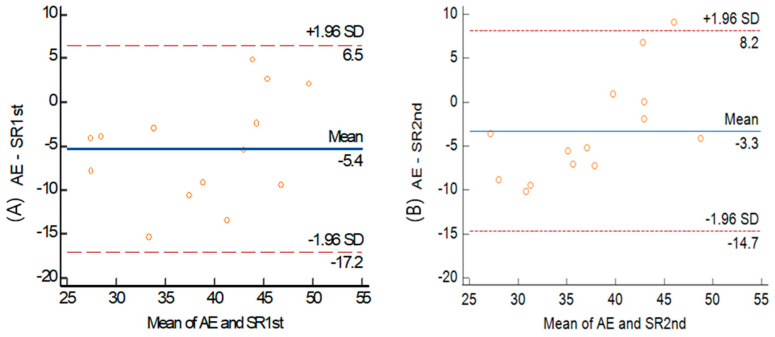
Bland–Altman plots of inter-method for VO_2_max (**A**) 1st shuttle ride (**B**) 2nd VO_2_max between arm ergometer and shuttle ride. Center solid line: mean difference (bias) between the two methods. Upper and lower lines: mean difference ± 1.96SD (95% limits of agreement).

**Figure 4 healthcare-12-00580-f004:**
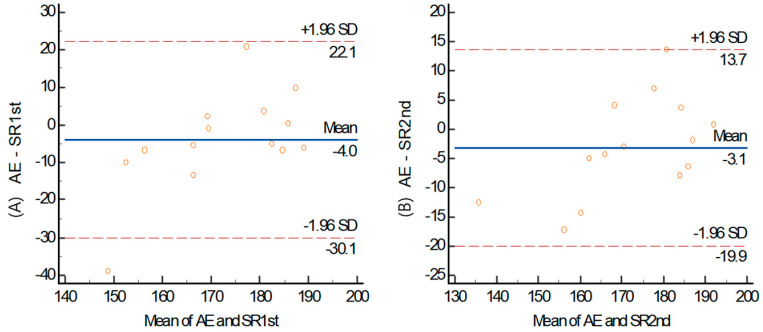
Bland–Altman plots of inter-method for HRmax (**A**) 1st shuttle ride (**B**) 2nd HRmax between arm ergometer and shuttle ride. Center solid line: mean difference (bias) between the two methods. Upper and lower lines: mean difference ± 1.96SD (95% limits of agreement).

**Figure 5 healthcare-12-00580-f005:**
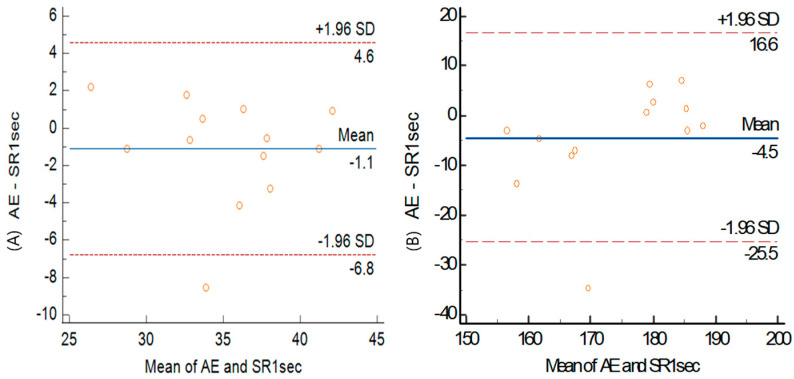
Bland–Altman plots of inter-method (**A**) VO_2_max (**B**) HRmax between arm ergometer and shuttle ride. Center solid line: mean difference (bias) between the two methods. Upper and lower lines: mean difference ± 1.96SD (95% limits of agreement).

**Table 1 healthcare-12-00580-t001:** The types and causes of disabilities.

Types of Disabilities		Causes of Disabilities	
Cerebral palsy	Locomotor disability	Congenital disability	Acquired disability
1	13	2	12

**Table 2 healthcare-12-00580-t002:** General characteristics of participants.

Sports	Variables	M ± SD
Para ice hockey (Experiment 1) (n = 14)	Experience (year)	12.1 ± 5.8
	Period of disability (year)	22.7 ± 11.6
	Disability classification	2.5 ± 1.16
	VO_2_max with AE (mL/kg/min)	35.9 ± 8.9
	HRmax with AE (beats/min)	170.6 ± 18.2
Wheelchair basketball (Experiment 2) (n = 13)	Experience (year)	10 ± 6
	Period of disability (year)	15.6 ± 7.5
	Disability classification	1.8 ± 1.0
	VO_2_max with AE (mL/kg/min)	34.6 ± 4.5
	HRmax with AE (beats/min)	171.8 ± 14.4

Values are mean ± standard deviation (S.D.).

**Table 3 healthcare-12-00580-t003:** Protocol for audio cues during shuttle rides.

Level	Shuttle Count	Distance (m)	Standard Audio Cue		+1 s		1.5 s	
			Time (s)	Speed (km/h)	Time (s)	Speed (km/h)	Time (s)	Speed (km/h)
1	7	15	8.5	6.4	9.5	5.7	10	5.4
2	8	15	8	6.8	9	6.0	9.5	5.7
3	8	15	7.6	7.1	8.6	6.3	9.1	5.9
4	9	15	7.2	7.5	8.2	6.6	8.7	6.2
5	9	15	6.9	7.8	7.9	6.8	8.4	6.4
6	10	15	6.5	8.3	7.5	7.2	8	6.8
7	10	15	6.3	8.6	7.3	7.4	7.8	6.9
8	11	15	6	9.0	7	7.7	7.5	7.2
9	11	15	5.8	9.3	6.8	7.9	7.3	7.4
10	11	15	5.5	9.8	6.5	8.3	7	7.7
11	12	15	5.3	10.2	6.3	8.6	6.8	7.9
12	12	15	5.1	10.6	6.1	8.9	6.6	8.2
13	13	15	5	10.8	6	9.0	6.5	8.3
14	13	15	4.8	11.3	5.8	9.3	6.3	8.6
15	13	15	4.6	11.7	5.6	9.6	6.1	8.9
16	14	15	4.5	12.0	5.5	9.8	6	9.0
17	14	15	4.4	12.3	5.4	10.0	5.9	9.2
18	15	15	4.2	12.9	5.2	10.4	5.7	9.5
19	15	15	4.1	13.2	5.1	10.6	5.6	9.6
20	16	15	1	13.5	5	10.8	5.5	9.8

## Data Availability

Data are contained within the article.

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
