# Peer review of "Development of Field Tests for Cardiovascular Fitness Assessment in Wheelchair"

_healthcare, 2024, doi:10.3390/healthcare12050580_

Round 1
Reviewer 1 Report
Comments and Suggestions for Authors
Cho et al. performed an elegant study. The manuscript is easy to follow and well describe a need of development of field tests for cardiovascular fitness assessment in wheelchair users. Study design is well designed and the results expressed throughly.
Authors did a good job and thank you for performing a nice study for wheelchair users.
My only minor comment is that the word " KPC athletes Village" sounds not appropriate. I suggest that the authors should consider using the word " KPC athletes training center". Additionally, the statistical power of the study is relatively small (n=14), but I understand that it is hard to recruit many paralympic athletes for this kind of test validation studies. Comments on the Quality of English LanguageManuscript is well written.
Reviewer 2 Report
Comments and Suggestions for Authors
The article entitled "Development of Field Tests for Cardiovascular Fitness Assessment in Wheelchair" provides relevant information for analysis in sport, facilitating access to crucial information for athlete development.
The introduction provides a good basis for what is proposed in the objectives and discussed throughout the text.
The methods are well described and make the whole procedure clear
The results are as expected and the discussion reinforces the importance of this analysis and of further work like this.
However, I would like to suggest some precautions and corrections:
Reduce the use of "etc...", adding all the variables included or creating a group of variables, for example, anthropometry (weight, height, fat mass...), injury (time, type, place...) so that the text is more fluid and easier to understand.
Line 103: The number of individuals recruited was 27, and after exclusions there were 27 left? Please explain how many were recruited, how many were excluded for specific reasons and how many were added to the analysis.
Line 130: When it comes to the test, in the case of this study I believe that showing with an image (even a photo of a participant if there is one) how the procedure was carried out can facilitate the reader's understanding, since as mentioned they are not traditional conditions of analysis for most researchers.
Lines 140 and 144: Why were the experiments divided into field hockey and basketball groups and not grouped together? Or even randomized? Is there a difference between the groups in terms of disability classification, which might not affect the results if evaluated separately?
Statistics: Our results and discussion section focuses only on the significance of the p-value. The p-values are acceptable, but it would be important to integrate the calculated effect sizes and confidence intervals into the results around the effect sizes.
Limitations of the study: I believe that the limitations can be softened (by reducing them to two paragraphs, for example). The paper is well written and can contribute to future research.
Reviewer 3 Report
Comments and Suggestions for Authors
Dear All,
I think what they are working on is an interesting topic. I'll leave some comments that are intended to improve what they have.
The introduction in general seems correct to me, I only have some doubts regarding the research problem. It is understood that the problem is that there is no test that is completely useful for athletes in wheelchairs, but I believe that they should expose the weaknesses or limitations of the tests that already exist, and the harm they cause to athletes. I believe that in this way the need to carry out the investigation would be justified.
Reading your objectives, I see that the first talks about validation and the second is to develop a field test. Here I have doubts with the first objective, since the introduction does not explain any of that (when reading it I thought that they would only create a new one). I think the test could be created and validated. Analyze and determine.
The second objective also says that the method is not limited by time, location or specialized equipment, I think that more information on these aspects should be added in the introduction.
In terms of material and method, the incorporated sample is clearly presented, but I believe that the inclusion/exclusion criteria should be highlighted to a greater extent (beyond those who dropped out during the process).
It explains how they measured VO2 directly and how people became familiar with the instruments.
Once again the question arises as to why they validate and then believe. They must work on it more so that it is better justified.
The biggest question I have is why the lines were 15 meters apart, and there was a 3 meter zone, along with the sound stimulus. I don't want to say that it is wrong, but that I think it is necessary to add information that justifies its use.
The results are well described.
The first paragraph of the discussion explains how reviews and meetings with experts were carried out. Which is very good, but I think that should be explained in more detail in the methodology.
Also, at the beginning of this chapter it says that the goal is to create a test, but you state two goals (validate and create). Modify where appropriate so that the information remains the same throughout the text.
The information presented in the discussion is good, but in my opinion, it should be incorporated in the introduction, since justifications are given for many of the comments I have written.
I think the discussion lacks greater comparison with similar research, work on that. They must also highlight the good things about their test beyond the significant values given by the statistics. They must explain what is new about their test, how it best adapts to the characteristics of the people for whom it was created, among others.
I hope my comments are a contribution.
Greetings.
Round 2
Reviewer 2 Report
Comments and Suggestions for Authors
The main purpose of the study was to revalidate established cardiovascular fitness assessment methods and to develop field tests for wheelchair athletes. The findings, derived from testing on nationally representative athletes, reveal that the 15m Shuttle Ride proves to be a suitable tool for assessing cardiovascular fitness in wheelchair athletes when validated against reference-based arm ergometry. Notably, the ultimate selection of an audio cue featuring a 1-second increase from the original Shuttle Run audio cue demonstrated stronger correlations in both VO2max and HRmax. The reviewers' recommendations were followed or clarified according to the requests. So, I recommend the publication of this manuscript in HEALTHCARE.
Reviewer 3 Report
Comments and Suggestions for Authors
Dear All,
I have reviewed your answers and they seem correct to me. I think the research is ready to be published.
Congratulations.